# Enterovirus 71 seroepidemiology in Taiwan in 2017 and comparison of those rates in 1997, 1999 and 2007

Jian-Te Lee[1], Ting-Yu Yen[2], Wei-Liang Shih[3], Chun-Yi Lu[2], Ding-Ping Liu[4,5], Yi-Chuan Huang[6], Luan-Yin Chang[2]*, Li-Min Huang[2], Tzou-Yien Lin[7,8]

1 Department of Pediatrics, National Taiwan University Hospital, Yun-Lin Branch, Yunlin, Taiwan,
2 Department of Pediatrics, National Taiwan University Hospital, National Taiwan University, Taipei, Taiwan,
3 Institute of Epidemiology and Preventive Medicine, College of Public Health, National Taiwan University and Infectious Diseases Research and Education Center, Ministry of Health and Welfare and National Taiwan University, Taipei, Taiwan, 4 Epidemic Intelligence Center, Centers for Disease Control, Taipei, Taiwan, 5 National Taipei University of Nursing and Health Sciences, Taipei, Taiwan, 6 Department of Pediatrics, Kaohsiung Chang Gung Memorial Hospital and Chang Gung University College of Medicine, Kaohsiung, Taiwan, 7 Department of Pediatrics, Chang Gung Memorial Hospital and Chang Gung University College of Medicine, Taoyuan, Taiwan, 8 The National Health Research Institutes, Miaoli, Taiwan

* lychang@ntu.edu.tw

**Data Availability Statement:** All relevant data are within the paper and its Supporting Information files.

## Abstract

### Background

During recent 20 years, enterovirus 71 (EV71) has emerged as a major concern among children, particularly in the Asia-Pacific region. To understand current EV71 serostatus, to find risk factors associated with EV71 infection and to establish future EV71 vaccine policy, we performed a seroepidemiology study in Taiwan in 2017.

### Methods

After informed consent was obtained, we enrolled preschool children, 6–15-year-old students, 16–50-year-old people. They received a questionnaire and a blood sample was collected to measure the EV71 neutralization antibody.

### Results

Altogether, 920 subjects were enrolled with a male-to-female ratio of 1.03. The EV71 seropositive rate was 10% (8/82) in infants, 4% (6/153) in 1-year-old children, 8% (7/83) in 2-year-old children, 8% (13/156) in 3–5-year-old children, 31% (38/122) in 6–11-year-old primary school students, 45% (54/121) in 12–15-year-old high school students and 75% (152/203) in 16-50-year-old people. Risk factors associated with EV71 seropositivity in preschool children were female gender, having siblings, more siblings, and contact with herpangina or hand-foot-and-mouth disease. The risk factor with EV71 seropositivity in 16–50-year-old people was having children in their families in addition to older age (p<0.001). Compared with the rates in 1997, 1999 and 2007, the rates in children were significantly lower in 2017.

**Funding:** This study was supported by grants from the Taiwan Centers for Disease Control, the Ministry of Health and Welfare, Taiwan (grant number MOHW 106-CDC-C-114-000117 to L-YC) and the Ministry of Science and Technology, Taiwan (grant numbers MOST 105-2320-B-002-016 and 105-2314-B-002-139-MY3) to L-YC. The funders had no role in study design, data collection and analysis, decision to publish, or preparation of the manuscript.

**Competing interests:** The authors have declared that no competing interests exist.

## Conclusion

EV71 seropositive rates were very low, at 4% to 10%, in preschool children and not high, at 31%, in primary school students. Preschool children are highly susceptible and need EV71 vaccine most.

## Introduction

Enterovirus 71 (EV71) was first isolated in California, USA in 1969 [1]. Since then, EV71 emerged in various regions throughout the world [2–7]. Large-scale outbreaks with frequent central nervous system (CNS)-complicated cases and deaths were found in Bulgaria, Hungary, Malaysia, Taiwan, Vietnam, Brunei, China and Cambodia [2–8]. An EV71 epidemic swept Taiwan in 1998, which caused 405 severe cases and 78 deaths [8–12]. After the epidemic, multiple and real-time national enterovirus surveillance systems were established by the Taiwan Centers for Disease Control, including viral lab network; outpatient, inpatient, and emergency room visits for hand-foot-and-mouth-disease (HFMD) and/or herpangina (HA); and mandatory notification of enterovirus severe cases [13–16]. After the first EV71 epidemic in 1998, EV71 cases occurred again in 2000–2001, 2005, 2008, and 2012 based on the surveillance data [15–17]. There seems to be a nationwide EV71 epidemic every 3 to 5 years. In addition to Taiwan, EV71 has emerged as a major concern among children in the Asia-Pacific region during recent 20 years [4,6,7].

After several epidemics, there is an urgent need to understand the current EV71 serostatus of high-risk groups and the high transmission population to predict future epidemics and to establish future EV71 vaccine policy [18]. In addition, we can compare the seroepidemiology in different years to determine the temporal change in the EV71 endemics in Taiwan. We thus performed this seroepidemiology study.

## Methods

### Study subjects

The Institutional Review Board of the National Taiwan University Hospital approved this study. After written informed consent was obtained from parents or guardians of children, we enrolled preschool children, 6–11-year-old primary school students and 12–15-year-old high school students in the northern (Taipei City), eastern (Hualien County), western (Yunlin County) and southern (Kaohsiung City) regions of Taiwan between May and November 2017. Community-dwelling 16-50-year-old healthy people were also enrolled in the four different regions of Taiwan after their own written informed consent was obtained. Participants received a questionnaire, and a blood sample was collected and submitted for measuring the EV71 neutralization antibody.

### Data collection for the serosurvey

The questionnaire solicited demographic data, residential area, number of children and adults in a family, vaccination history, past history of HFMD and/or HA, intrafamilial or outside contact with HFMD and/or HA cases, family members with HFMD and/or HA, classmates or neighbors with HFMD and/or HA, sources of drinking water, employment of a babysitter, enrollment in a kindergarten or childcare center, and breastfeeding during infancy. No EV71 vaccine is licensed in Taiwan up to now. Contact with HFMD and/or HA cases was defined as

kissing, hugging, shaking hands, sharing food, or playing with children who had HFMD and/ or HA. All interviewers were trained, and contact history information was collected from several family members to minimize recall bias. The questionnaires for preschool children, students and adults are listed in the supplementary files (S1, S2 and S3 Files).

## Laboratory methods for the EV71 neutralizing antibody

The serum neutralizing antibody response is the major indicator of EV71 protective immunity. The neutralizing antibody test of EV71 followed the standard protocol of a neutralization test. Serum samples were heat-treated for 30 minutes at 56˚C, serially diluted and mixed with 100 50% tissue culture-infective doses (TCID50) of the EV71 TW/2272/98 strain (GenBank accession number AF119795), and incubated for 2 hours at 37˚C. Thereafter, rhabdomyosarcoma cells were added into each reaction well and incubated at 37˚C in 5% $CO_2$ incubator. Each plate included a cell control, serum control, and virus back-titration. The cytopathic effect was monitored from 5 to 6 days after incubation, and the serotiter was determined when the cytopathic effect was observed in one TCID50 of the virus back-titration. Seropositivity was defined as a serotiter ≥8. For details, please see http://dx.doi.org/10.17504/protocols.io. 7pwhmpe.

## Comparison of EV71 serostatus among 1997, 1999, 2007 and 2017

We previously performed EV71 seroepidemiology studies in 1997, 1999, and 2007, which were conducted by Dr. Luan-Yin Chang and Dr. Tzou-Yien Lin [19]. The method for neutralization antibody measurement was the same. We thus compared the age-specific seropositive rates among different years.

## Statistical analyses

We analyzed the data with the SAS statistical software (version, SAS Institute, Cary, North Carolina). We used Student's *t*-test for continuous data and chi-squared tests for categorical data. A p value <0.05 indicated a statistical significance.

## Results

### Demography and EV71 serostatus in 2017

We conducted nationwide recruitment from urban and rural regions, as Table 1 shows. Taipei City and Kaohsiung City, two metropolitan areas, are located in the northern and southern regions of Taiwan, whereas Hualien County and Yunlin County, two rural areas, are located in the eastern and western regions of Taiwan.

In total, 920 subjects were enrolled with a male-to-female ratio of 1.03. Direct standardization method was applied to calculate age-region standardized seroprevalence for male and female, using the population of Taipei City, Hualien County, Yunlin County and Kaohsiung City in 2017 as the standard population. Overall, females had significantly greater seropositive rates than males, as Fig 1 shows (p = 0.01). The EV71 seropositive rate was 10% (8/82) in infants, 4% (6/153) in 1-year-old children, 8% (7/83) in 2-year-old children, 8% (13/156) in 3–-5-year-old children, 31% (38/122) in 6–11-year-old primary school students, 45% (54/121) in 12–15-year-old high school students, 80% (97/122) in 16-50-year-old women and 68% (55/81) in 16-50-year-old men in 2017.

There was significant difference of seropositive rates among preschool and primary student children between the rural areas and metropolitan areas, as Table 1 shows. The seropositive rates of 3–5-year-old kindergarteners and 6–11-year-old primary school children were

**Table 1. Age-specific EV71 seropositive rates in four different regions of Taiwan in 2017.**

| Age in years | Overall | Taipei (N) | Kaohsiung (S) | Yunlin (W) | Hualien (E) | p value |
|---|---|---|---|---|---|---|
| <1 | 10% (8/82) | 10% (3/29) | 5% (1/21) | 16% (4/25) | 0% (0/7) | 0.48 |
| 1 | 4% (6/153) | 10% (3/31) | 2% (1/45) | 0% (0/32) | 4% (2/45) | 0.22 |
| 2 | 8% (7/83) | 14% (4/28) | 0% (0/9) | 4% (1/23) | 9% (2/23) | 0.46 |
| 3–5 | 8% (13/156) | 4% (3/70) | 0% (0/26) | 14% (4/29) | 19% (6/31) | 0.02 |
| 6–11 | 31% (38/122) | 20% (6/30) | 21% (7/34) | 37% (10/27) | 48% (15/31) | 0.04 |
| 12–15 | 45% (54/121) | 26% (8/31) | 54% (15/28) | 45% (14/31) | 55% (17/31) | 0.08 |
| 16–50 | 75% (152/203) | 73% (37/51) | 63% (32/51) | 84% (42/50) | 80% (41/50) | 0.07 |

The p value was measured by chi-squared test. N denotes the northern region, W is the western region, S is the southern region, and E is the eastern region. Taipei City and Kaohsiung City are metropolitan areas, whereas Hualien County and Yunlin County are rural areas.

significantly greater in rural areas (Hualien County and Yunlin County) than in metropolitan areas (Taipei City and Kaohsiung City). Among the four regions, there were no significant differences in the age and gender of the enrolled people, but the family size was larger (4.4±1.8 vs. 3.9±1.7, p<0.001) and the rate of drinking spring or well water was significantly greater (18% vs. 9%, p<0.05) in rural areas than in metropolitan areas.

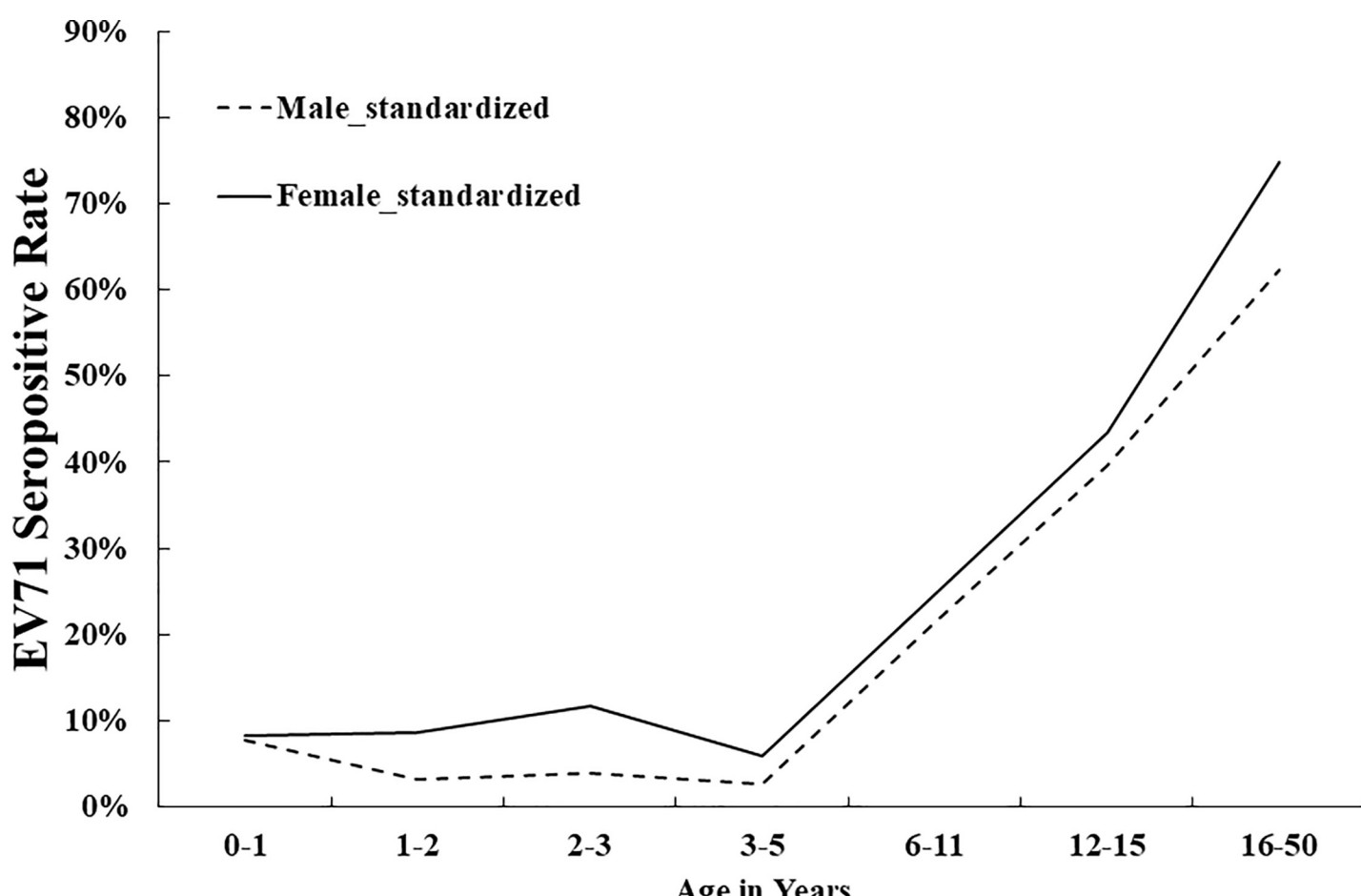

**Fig 1. Age-specific EV71 serostatus between males and females in 2017.** Overall, females had significantly greater seropositive rates than males (p = 0.01).

**Table 2. Risk factors associated with EV71 seropositivity in preschool children in 2017.**

| Factor | | Seropositive (N = 34) | Seronegative (N = 440) | OR, 95% CI | p value |
|---|---|---|---|---|---|
| **Sex** | Male | 13 (38.2%) | 245 (55.7%) | Ref. | |
| | Female | 21 (61.8%) | 195 (44.3%) | 2.03, (1.00, 4.26) | 0.05 |
| **Age (years)** | Mean±SD | 2.8±1.8 | 2.5±1.6 | | 0.45 |
| **Residential area** | City | 23 (67.6%) | 287 (65.2%) | Ref. | |
| | Village | 9 (26.5%) | 113 (25.7%) | 0.45, (0.45, 2.21) | 0.99 |
| | Suburban | 2 (5.9%) | 35 (8.0%) | 0.71, (0.16, 3.15) | 0.66 |
| | Industrial | 0 (0%) | 5 (1.1%) | NA | |
| **Household size** | Mean±SD | 4.1±1.7 | 4.3±1.8 | | 0.53 |
| **Sibling** | No | 6 (17.7%) | 167 (38.0%) | Ref. | |
| | Yes | 28 (82.3%) | 273 (62.0%) | 2.85, (1.16, 7.04) | 0.02 |
| | Mean±SD | 1.2±1.0 | 0.8±0.8 | | 0.005 |
| **History of chronic illness** | No | 29 (85.3%) | 404 (91.8%) | Ref. | |
| | Yes | 5 (14.7%) | 36 (8.2%) | 1.93, (0.71, 5.30) | 0.19 |
| **Ever breastfed** | No | 1 (2.9%) | 31 (7.1%) | Ref. | |
| | Yes | 33 (97.1%) | 409 (92.9%) | 2.50, (0.33, 18.89) | 0.36 |
| **Attendance of kindergartens or daycare center** | No | 9 (26.5%) | 163 (37.1%) | Ref. | |
| | Yes | 25 (73.5%) | 277 (62.9%) | 1.64, (0.75, 3.59) | 0.22 |
| **Main source of drinking water** | Tap water | 19 (55.9%) | 260 (59.1%) | Ref. | |
| | Well water | 0 (0%) | 2 (0.5%) | NA | |
| | Spring | 4 (11.8%) | 51 (11.5%) | 1.07, (0.35, 3.29) | 0.90 |
| | Others | 11 (32.3%) | 127 (28.9%) | 1.19, (0.55, 2.57) | 0.67 |
| **History of HFMD** | No | 22 (64.7%) | 334 (75.9%) | Ref. | |
| | Yes | 11 (32.4%) | 99 (22.5%) | 1.69, (0.79, 3.60) | 0.18 |
| | Unknown | 1 (2.9%) | 7 (1.6%) | | |
| **History of herpangina** | No | 26 (76.5%) | 316 (71.8%) | Ref. | |
| | Yes | 8 (23.5%) | 110 (25.0%) | 0.88, (0.39, 2.01) | 0.77 |
| | Unknown | 0 (0.0%) | 14 (3.2%) | | |
| **Family member with history of HFMD and/or herpangina** | No | 20 (58.8%) | 288 (65.5%) | Ref. | |
| | Yes | 13 (38.2%) | 134 (30.4%) | 1.40, (0.68, 2.89) | 0.37 |
| | Unknown | 1 (2.9%) | 18 (4.1%) | | |
| **Classmates with history of HFMD and/or herpangina** | No | 1 (2.9%) | 38 (8.6%) | Ref. | |
| | Yes | 19 (55.9%) | 134 (30.5%) | 5.39, (0.70, 41.56) | 0.11 |
| | Unknown | 14 (41.2%) | 268 (60.9%) | | |
| **Contact with history of HFMD or herpangina** | No | 7 (20.6%) | 173 (39.3%) | Ref. | |
| | Yes | 18 (52.9%) | 144 (32.7%) | 3.09, (1.26, 7.60) | 0.01 |
| | Unknown | 9 (26.5%) | 123 (28.0%) | | |

The p values were measured by Student's t-test or chi-squared test. SD denotes standard deviation; HFMD, hand, foot and mouth disease; OR, odds ratio; CI, confidence interval; NA, not available.

## Risk factors associated with EV71 seropositivity

Table 2 shows risk factors associated with EV71 seropositivity in preschool children aged less than 6 years old and significant risk factors were female gender, having siblings, more siblings, and contact with HFMD and/or HA. Residential areas, breast feeding, vaccination history, and source of drinking water did not affect the EV71 seropositivity in preschool children.

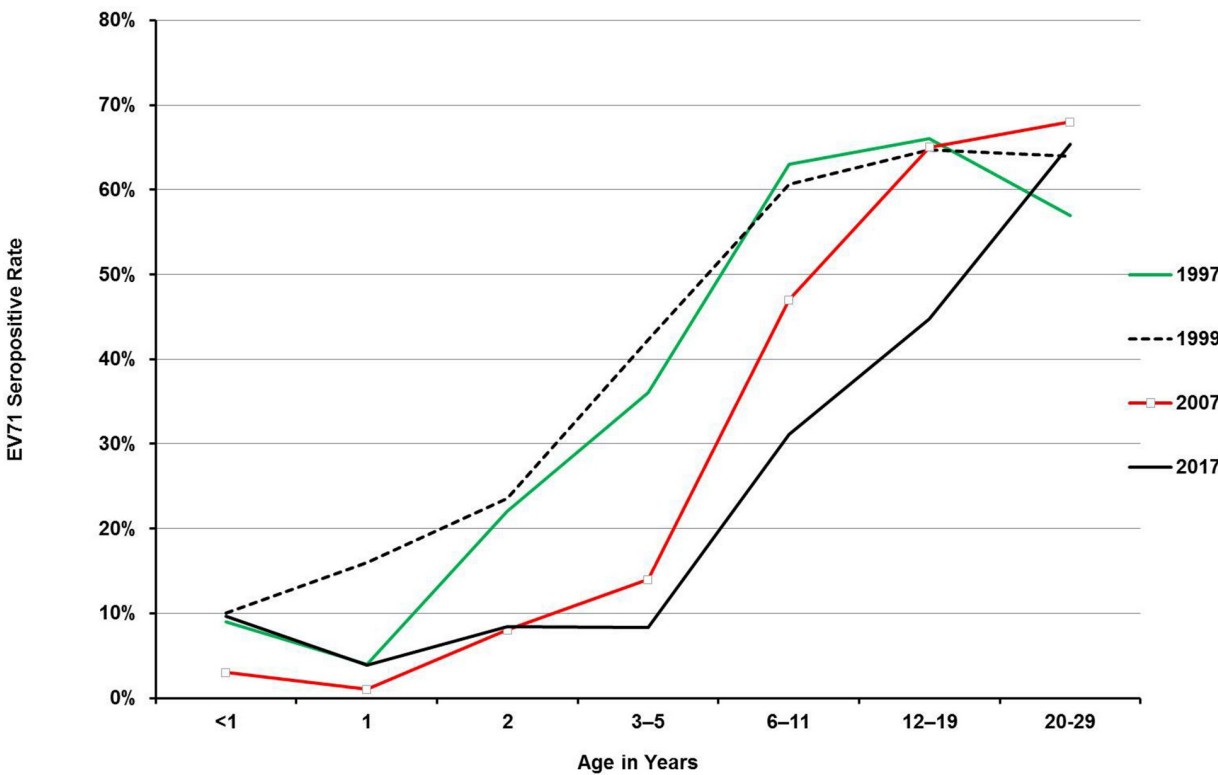

**Fig 2. Age-specific EV71 serostatus in Taiwan in 1997, 1999, 2007, and 2017.** Infants did not have significantly different seropositive rates among different years (p = 0.62). The rates in 2–11-year-old children were significantly lower in 2007 and 2017 than those rates in 1997 and 1999 (p<0.001). The seropositive rate of the 6- to 19-year-olds in 2017 was significantly lower than those rates in 1997, 1999 and 2007 (p<0.001). The seropositive rates of 20- to 29-year-old adults were not significantly different among the 4 different years.

For the 6 to 15 year-old children, no significant risk factors were found except older age. For 16-50-year-old people, the significant risk factor associated with EV71 seropositivity was having children in addition to older age: 61% of seropositive 16-50-year-old people had children in their families but only 27% of seronegative 16-50-year-old people had children in their families (p<0.001).

## Comparison of EV71 serostatus among 1997, 1999, 2007 and 2017

Fig 2 shows age-specific seropositive rates among 1997, 1999, 2007 and 2017. Infants might have maternal antibodies, so they did not have significantly different seropositive rates among different years (p = 0.62). Compared with EV71 seropositive rates in 1997 and 1999, the rates in 2–11-year-old children were significantly lower in 2007 and 2017 as Fig 2 shows (p<0.001). The seropositive rates of the 6- to 19-year-olds in 2017 were significantly lower than those rates in 1997, 1999 and 2007 (p<0.001). There were no significant differences in 20- to 29-year-old adults among the four different periods (p = 0.36). Overall, the seropositive rates in 2017 were the lowest among the four periods.

## Discussion

This EV71 seroepidemiology study in 2017 shows that seropositive rates in children were significantly lower in comparison with those rates in 1999 [19,20], one year after the first epidemic occurred in Taiwan in 1998. The EV71 seropositive rates in 2017 were very low, at 4%

to 10%, in the very-high-risk group (<6-year-old preschool children) and not high, at 31%, in primary school students. The risk factors associated with seropositivity in preschool children included female gender, having siblings, more siblings, and contact with HFMD and/or HA whereas the only risk factor associated with seropositivity in adults was having children in the family in addition to older age.

The lower seropositive rates in young children observed in 2007 and 2017 correlate well with the fact that there were no major EV71 outbreaks during the last 10 years. Factors underlying the decreasing seropositivity may include lower birth rates and strict implementation of preventive measures during this time period [21]. The low birth rate and having only one child in most families in Taiwan make children get EV71 infection at an older age in comparison with the previous periods of the late 1990s or early 2000s. In our previous study conducted in 2001–2002, we found a very high EV71 household transmission rate of 84% among siblings if there were any siblings infected with EV71 in the household [22]. Taiwan has the sixth-lowest birth rate in the world, according to the most recent data released by the Central Intelligence Agent World Factbook [23], and the fertility rate in Taiwan was only 1.13 children per woman in 2017 and 1.218 children per woman in 2018, which was the third lowest in the world [23]. Because of having only one child in most families in recent years, there is no elder sibling(s) to spread viruses, including EV71, into the other family members; thus, most children acquire viral infection when they go to school or kindergarten. Therefore, the age of getting EV71 becomes older than in previous periods. In addition, EV71 infection at different ages has different symptomatic ratios and severity [19]. The symptomatic ratios are much lower [19], and the severity is milder when children get an EV71 infection at older ages [13, 24].

After the nationwide EV71 epidemic that occurred in Taiwan in 1998, multiple and real-time national enterovirus surveillance systems have been established by the Taiwan Centers for Disease Control since 2000 and include viral lab network; outpatient, inpatient, and emergency room visits for HFMD and/or HA from National Health Insurance claim data; and mandatory notification of enterovirus severe cases [13–16, 21]. The real-time EV surveillance makes early detection of EV71 possible, and strict implementation of preventive measures during EV71 circulation periods are applied, so these preventative measures may limit the spread of EV71 [21]. For example, to reduce the risk of enterovirus clustering, class suspension has been executed for the pre-school education and care institutions. Strengthened implementation of infection control measures is applied in hospitals and postpartum nursing care centers to reduce the risk of enterovirus clusters. For medical care of severe enterovirus cases, Taiwan Centers for Disease Control have constructed a medical service network for recent 2 decades and workshops are held on the clinical treatment of critical enterovirus complications to enhance doctors' skills every year. There has been a marked decrease in severe and fatal EV71 cases after 2012 [17], and the results of this seroepidemiology study also endorse the impact of a well-established real-time enterovirus surveillance system and all the preventive measures to limit the spread of EV71 in Taiwan. Other speculations may include viral evolution, as evidenced by EV71-like gene elements appearing in other cocirculating EV-A viruses by intensive recombination [25]. The virulence of EV71 could thus be modified. Overall, the viral (recombination and mutation), host (low birth rate) and environmental (increased surveillance) factors might change transmission dynamics of EV71. However, with the accumulation of a susceptible high-risk group during the past 20 years, new epidemics could occur in the years to come. Careful monitoring and preventive measures are still necessary.

Risk factors associated with EV71 seropositivity were female gender, having siblings, more siblings, classmates with HFMD and/or HA and contact with HFMD and/or HA in high-risk groups. The seropositive rate in females is significantly greater than in males. However, the male-to-female ratio of symptomatic and/or severe enterovirus infections shifted toward

males and was approximately 1.5 [10,13,24]. The role of gender on the susceptibility to EV71 and the severity of EV71 infections remains unclear and needs further investigation.

Among different regions in Taiwan, 3- to 5-year-old children and primary school students in rural areas had greater seropositive rates. In our previous seroepidemiology conducted in 1999, living in rural areas was also a significant risk factor for EV71 infection [19]. The possible reasons may be poorer hygiene behaviors and larger family size, resulting in more possible contacts and transmissions of EV71 in rural areas. The socioeconomic and educational gaps between urban and rural areas might affect the parents' and/or caregivers' hygiene behavior, which would in turn have some impact on the infection incidence and/or severity. For example, caregivers in the critical group (encephalitis with or without cardiopulmonary failure) of EV71 infection had a significantly lower rate in terms of cleaning the faucet after washing their hands, as our previous study reported [24].

Young children in Taiwan have very low EV71 seropositive rates now and therefore harbor a high risk of developing an EV71 infection. Our earlier studies also showed that young children are at especially high risk of developing severe disease after EV71 infections [24]. Future EV71 vaccine policy should be targeted to young children, who will be the first priority to receive the EV71 vaccine in Taiwan.

For 16-50-year-old people, the significant risk factor associated with EV71 seropositivity was having children in addition to older age. Bidirectional transmission of EV71 between children and parents or other household adults does occur, parents or other household adults might transmit EV71 to their children and vice versa. For example, a Japanese mother developed encephalitis caused by intrafamilial transmission of EV71 from her 1-year-old son and it also elucidated the risk for EV71 encephalitis even in adults [26]. Therefore, parents and other household adults have to be cautious and take preventive measures once their children have such infection.

We further compared EV71 serostatus among different countries in Table 3. Among different countries, the EV71 seropositive rates varied greatly. The seropositive rates of children in China, Thailand and Cambodia were very high, ranging from 35 to 98% [27–31]. The rates (15 to 37%) of children in Singapore in 2008–2010 were comparable to those in Taiwan in 2007 and 2017 [28]. The rates in Russia varied in different regions: the rates were lower (5 to 19%) in urban areas, such as Moscow and Rostov, than those (20 to 83%) in rural areas, such as Tyva [29]. Severe EV71 cases occurred more frequently in countries, such as China and Cambodia which had greater seropositive rates [7, 23, 31–33] and it indicates greater disease burden in these countries. EV71 outbreak occurred in Cambodia in 2012, it characterized by severe encephalitis with cardiovascular collapse and pulmonary edema seized international headlines and resulted in the death of at least 54 children [31], and more than 2 thousands of laboratory-confirmed EV71 cases died in China from 2008 to 2015 [7].

There are some limitations in this study. First, the test population (sampling methods, geographical and demographical characteristics) were not the same with those previous studies of 1997, 1999 and 2007 in Taiwan although the laboratory method was the same. Second, the sample size of young children was small and the power may be limited, so larger sample size and multiple test correction may be needed in the future to verify our findings. Third, the EV71 viral strains used for neutralizing antibody, cut-off criteriae for seropositivity and the age groups are different among different countries. Although the seroprevalence rates were not so comparable, we tried our best to clarify the difference and did some important comparisons.

## Conclusions

The EV71 seropositive rates were very low, at 4% to 10%, in the high-risk group (<6-year-old preschool children) and not high, at 31%, in primary school students. Female gender, having

**Table 3. Comparison of the age-specific EV71 serostatus among different countries.**

| | Country and Year | | | | | | | | | |
|---|---|---|---|---|---|---|---|---|---|---|
| | Taiwan | Taiwan | Taiwan | Taiwan | Guangzhou, China | Singapore | Russia | Thailand | Cambodia | Cambodia |
| Age | 1997 | 1999 | 2007 | 2017 | 2014–2015 | 2008–2010 | 2007–2008 | 2009–2012 | 2000–2005 | 2006–2011 |
| <1 | 9% (5/56) | 10% (60/589) | 3% (1/32) | 10% (8/82) | 35% [a] (6/17) | NA | NA | 43% [d] (17/40) | NA | NA |
| 1 | 4% (2/50) | 16% (65/407) | 1% (1/89) | 4% 6/153) | | | | | NA | NA |
| 2 | 22% (4/19) | 24% (86/365) | 8% (9/118) | 8% (7/83) | 43% (26/60) | 15% [b] (16/104)) | 5–20% [c] (3/67-11/54) | | 52% | 93% |
| 3–5 | 36% (9/25) | 42% (309/730) | 14% (42/311) | 8% (13/156) | 71% (85/120) | | 19–83% (16/82-70/84) | 48% (14/29) | 52–76% | 93–97% |
| 6–11 | 63% (18/29) | 61% (461/761) | 47% (338/717) | 31% (38/122) | NA | 26% (77/294) | NA | 88% (30/34) | 50% [e] | 88–98% |
| 12–19 | 66% (51/78) | 65% (419/648) | 65% (459/705) | 46% (60/134) | NA | 37% (112/302) | NA | 80% (24/30) | NA | 86–98% [f] |
| 20–29 | 57% (29/51) | 64% (229/358) | 68% (339/500) | 65% (49/75) | NA | NA | NA | 77% (17/22) | NA | NA |

The cutoff for EV71 seropositivity was ≥1:8 [19, 27–30], except in the Cambodia study, which used a titer of ≥1:16 as the cutoff [31].

NA: not available.

Numbers in parenthesis are case number with EV71 seropositivity/case number tested but those numbers are not available in Cambodia.

[a]The rate for children younger than 2 years [27].

[b]The rate for 1–6 year-old children [28].

[c]The rate for 1–2 year-old children [29].

[d]The rate for 6 month-old to 2 year-old children [30].

[e]The rate for 6 year-old children [31].

[f]The rate for 12 to 15 year-old adolescents [31].

siblings, more siblings and contact with HA or HFMD were factors associated with EV71 seropositivity in children younger than 6 years of age while having children in the family and older age were factors associated with EV71 seropositivity in 16-50-year-old people. Young children will be the group with the greatest priority to receive the EV71 vaccine.

## Supporting information

**S1 File. Questionnaire for preschool children with Chinese-English parallel texts.**
(PDF)

**S2 File. Questionnaire for students with Chinese-English parallel texts.**
(PDF)

**S3 File. Questionnaire for adults with Chinese-English parallel texts.**
(PDF)

## Author Contributions

**Conceptualization:** Luan-Yin Chang.

**Data curation:** Jian-Te Lee, Ting-Yu Yen, Wei-Liang Shih, Yi-Chuan Huang, Tzou-Yien Lin.

**Formal analysis:** Wei-Liang Shih, Luan-Yin Chang.

**Funding acquisition:** Ding-Ping Liu, Luan-Yin Chang.

**Investigation:** Ting-Yu Yen.

**Methodology:** Ding-Ping Liu.

**Resources:** Luan-Yin Chang, Li-Min Huang.

**Supervision:** Chun-Yi Lu, Luan-Yin Chang, Li-Min Huang, Tzou-Yien Lin.

**Validation:** Wei-Liang Shih, Chun-Yi Lu, Ding-Ping Liu, Li-Min Huang, Tzou-Yien Lin.

**Writing – original draft:** Jian-Te Lee, Luan-Yin Chang.

**Writing – review & editing:** Ting-Yu Yen, Wei-Liang Shih, Chun-Yi Lu, Ding-Ping Liu, Yi-Chuan Huang, Luan-Yin Chang, Li-Min Huang, Tzou-Yien Lin.

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
