## [Decision Letter · Decision Letter 0]

10 Sep 2019

[EXSCINDED]

PONE-D-19-22806

Enterovirus 71 Seroepidemiology in Taiwan in 2017 and comparison of those rates in 1997, 1999 and 2007

PLOS ONE

Dear Prof. Chang,

Thank you for submitting your manuscript to PLOS ONE. Your manuscript has been reviewed by two experts in the field and their comments follow. While both are supportive, they have also raised concerns that you should address in a revised manuscript.

After careful consideration, we feel that your paper has merit and we invite you to submit a revised version of the manuscript.

We would appreciate receiving your revised manuscript by Oct 25 2019 11:59PM. To enhance the reproducibility of your results, we recommend that if applicable you deposit your laboratory protocols in protocols.io, where a protocol can be assigned its own identifier (DOI) such that it can be cited independently in the future. For instructions see: http://journals.plos.org/plosone/s/submission-guidelines#loc-laboratory-protocols

We look forward to receiving your revised manuscript.

Kind regards,

Dong-Yan Jin

Academic Editor

PLOS ONE

Journal Requirements:

1. Please ensure that your manuscript meets PLOS ONE's style requirements, including those for file naming. The PLOS ONE style templates can be found athttp://www.journals.plos.org/plosone/s/file?id=wjVg/PLOSOne_formatting_sample_main_body.pdf and http://www.journals.plos.org/plosone/s/file?id=ba62 /PLOSOne_formatting_sample_title_authors_affiliations.pdf

The name of the colleague or the details of the professional service that edited your manuscriptA copy of your manuscript showing your changes by either highlighting them or using track changes (uploaded as a *supporting information* file)A clean copy of the edited manuscript (uploaded as the new *manuscript* file

3. We noticed you have some minor occurrence of overlapping text with the following previous publication, which needs to be addressed:

- https://jbiomedsci.biomedcentral.com/articles/10.1186/s12929-019-0552-7

In your revision ensure you cite all your sources (including your own works), and quote or rephrase any duplicated text outside the methods section. Further consideration is dependent on these concerns being addressed.

Additionally, please address the following:

- whether a prior sample size calculation was carried out for this study

- the rationale for the lack of multiple test correction in your statistical analysis

- ensure you include details concerning the pretesting of your questionnaire included in this study and also a copy of it

Additional Editor Comments:

Please respond to reviewers' comments and provide the requested information if available.

Reviewers' comments:

Reviewer's Responses to Questions

**Comments to the Author**

1. Is the manuscript technically sound, and do the data support the conclusions?

Reviewer #1: Yes

Reviewer #2: Partly

2. Has the statistical analysis been performed appropriately and rigorously? 

Reviewer #1: I Don't Know

Reviewer #2: Yes

3. Have the authors made all data underlying the findings in their manuscript fully available?

Reviewer #1: Yes

Reviewer #2: Yes

4. Is the manuscript presented in an intelligible fashion and written in standard English?

Reviewer #1: Yes

Reviewer #2: Yes

5. Review Comments to the Author

Reviewer #1: The manuscript “Enterovirus 71 Seroepidemiology in Taiwan in 2017 and comparison of those rates in 1997, 1999 and 2007” provided an updated serostatus of EV71 in different region of Taiwan. This is a timely study as the threat of enterovirus infection continuously posing significant concern worldwide.

Below are my comments for improvement.

1) Is there any patient data co-relating with the seropositive specimen? For example, successful rate of detecting viral RNAs/antigens in specimens of seropositive patients; the history of seropositive patients with enterovirus-related syndromes.

2) The authors suggested that female gender could be a risk factor of EV71 seropositivity. The comparison appears to be based on the absolute number of seropositivity in between male and female. The comparison may be subjected to the biases due to disproportion of male:female ratio within the community. The authors may need to consider this point and perform normalization accordingly.

3) The number of tested specimen (n) needs to be indicated in table 3.

4) The authors suggested the lower EV71 seropositivity detected in specimen of 2007 and 2017 could be due to the absence of disease outbreak. Does the high seropositivity detected in other countries, for example, Cambodia (Table 3), correlate to an EV71 outbreak?

5) Figure 2 showed similar but delayed trend of EV71 seropositivity between specimens obtained from 1997 and 1999 to 2007 and 2017. For example, an age 1 EV71 seropositive specimen detected in 1999 will become an age 8 EV71 seropositive specimen detected in 2007. In other words, the delay could be an indication of effective infectious disease control. The authors should discuss what infectious disease control policy may have been implemented during this period of time. When comparing with data from other countries, which ways do Taiwan do better or worse?

Reviewer #2: Comments to the Author

The manuscript aims to explore the EV71 serostatus in Taiwan. The authors made efforts to test the EV71 neutralizing antibody with serum samples and compared EV71 serostatus among 1997, 1999, 2007 and 2017. They illustrated that the EV71 seropositive rates in children were significantly lower in 2017, compared with the rates in 1997, 1999 and 2008. These results prompt the necessity of continuous surveillance for the prevention and control of the disease infected with EV71. The main limitations of the manuscript are listed below.

1. Did the questionnaire include the subjects' vaccination history?

2. As the authors indicated that the male-to-femal ratio of symptomatic and/or severe enterovirus infections shifted toward males in the discussion section. While the female gender was the risk factors associated with EV71 seropositivity in this study, and this conclusion was based the analysis of in preschool children in 2017. The risk factors could be analyzed including other ages.

6. PLOS authors have the option to publish the peer review history of their article (what does this mean?). If published, this will include your full peer review and any attached files.

Reviewer #1: Yes: Man Lung YEUNG

Reviewer #2: No

---

## [Author Response · Author response to Decision Letter 0]

3 Oct 2019

Reviewers' comments:

Comments to the Author

Reviewer #1: The manuscript “Enterovirus 71 Seroepidemiology in Taiwan in 2017 and comparison of those rates in 1997, 1999 and 2007” provided an updated serostatus of EV71 in different region of Taiwan. This is a timely study as the threat of enterovirus infection continuously posing significant concern worldwide.

Response: thank you very much for your comment and appreciation.

Below are my comments for improvement.

1) Is there any patient data co-relating with the seropositive specimen? For example, successful rate of detecting viral RNAs/antigens in specimens of seropositive patients; the history of seropositive patients with enterovirus-related syndromes.

Response: we did not have any data of detecting viral RNAs/antigens in specimens of seropositive patients, but we did have data on enterovirus-related ( HFMD or herpangina) history of seropositive children in Table 2. For example, 34% of seropositive preschool children had history of HFMD and 24% had history of herpangina (Table 2).

2) The authors suggested that female gender could be a risk factor of EV71 seropositivity. The comparison appears to be based on the absolute number of seropositivity in between male and female. The comparison may be subjected to the biases due to disproportion of male:female ratio within the community. The authors may need to consider this point and perform normalization accordingly.

Response: we agree with this comment, so normalization was done for Figure 1 accordingly to minimize any disproportion of male:female ratio within the community. 

3) The number of tested specimen (n) needs to be indicated in table 3.

Response: Thank you very much for your suggestion and we added the numbers of seropositive and tested specimen in Table 3. However, only those numbers are not available in Cambodia and we can not add them.

4) The authors suggested the lower EV71 seropositivity detected in specimen of 2007 and 2017 could be due to the absence of disease outbreak. Does the high seropositivity detected in other countries, for example, Cambodia (Table 3), correlate to an EV71 outbreak?

Response: We agree with your comment that the high seropositivity detected in other countries correlate to an EV71 outbreak. We mentioned it in the Discussion: Severe EV71 cases occurred more frequently in countries, such as China and Cambodia which had greater seropositive rates [7, 23, 31-33] and it indicates greater disease burden in these countries. EV71 outbreak occurred in Cambodia in 2012, it characterized by severe encephalitis with cardiovascular collapse and pulmonary edema seized international headlines and resulted in the death of at least 54 children [31], and more than 2 thousands of laboratory-confirmed EV71 cases died in China from 2008 to 2015 [7].

5) Figure 2 showed similar but delayed trend of EV71 seropositivity between specimens obtained from 1997 and 1999 to 2007 and 2017. For example, an age 1 EV71 seropositive specimen detected in 1999 will become an age 8 EV71 seropositive specimen detected in 2007. In other words, the delay could be an indication of effective infectious disease control. The authors should discuss what infectious disease control policy may have been implemented during this period of time. When comparing with data from other countries, which ways do Taiwan do better or worse?

Response: Thank you very much for your detailed review and suggestion. We described more infectious disease control policy in the Discussion to share our experience, which may be helpful to other countries.

Reviewer #2: Comments to the Author

The manuscript aims to explore the EV71 serostatus in Taiwan. The authors made efforts to test the EV71 neutralizing antibody with serum samples and compared EV71 serostatus among 1997, 1999, 2007 and 2017. They illustrated that the EV71 seropositive rates in children were significantly lower in 2017, compared with the rates in 1997, 1999 and 2008. These results prompt the necessity of continuous surveillance for the prevention and control of the disease infected with EV71. The main limitations of the manuscript are listed below.

1. Did the questionnaire include the subjects' vaccination history?

Response: The questionnaire include the subjects' vaccination history. We deposited the questionnaires as supplementary files with Chinese-English parallel texts. We did not show it in the original manuscript since EV71 vaccine is not commercially available now in Taiwan and the current vaccination did not affect EV71 serostatus. We added it in the revised manuscript since you and the readers may be curious about it.

2. As the authors indicated that the male-to-female ratio of symptomatic and/or severe enterovirus infections shifted toward males in the discussion section. While the female gender was the risk factors associated with EV71 seropositivity in this study, and this conclusion was based the analysis of in preschool children in 2017. The risk factors could be analyzed including other ages.

Response: Thank you very much for your suggestion. We analyzed the risk factors in the other age groups, which are shown in the text: For the 6 to 15 year-old children, no significant risk factors were found except older age. For 16-50-year-old people, the significant risk factor associated with EV71 seropositivity was having children in addition to older age: 61% of seropositive 16-50-year-old people had children in their families but only 27% of seronegative 16-50-year-old people had children in their families (p<0.001). Therefore, we did not find gender difference in the other age groups.

---

## [Editor Report · Decision Letter 1]

7 Oct 2019

Enterovirus 71 Seroepidemiology in Taiwan in 2017 and comparison of those rates in 1997, 1999 and 2007

PONE-D-19-22806R1

Dear Dr. Chang,

Thank you for submitting your revised manuscript. I have read it through carefully together with your response to reviewers' comments. I am fully satisfied with the modifications you made.

We are pleased to inform you that your manuscript has been judged scientifically suitable for publication and will be formally accepted for publication once it complies with all outstanding technical requirements.

With kind regards,

Dong-Yan Jin

Academic Editor

PLOS ONE
---

## [Editor Report · Acceptance letter]

9 Oct 2019

PONE-D-19-22806R1 

Enterovirus 71 Seroepidemiology in Taiwan in 2017 and comparison of those rates in 1997, 1999 and 2007 

Dear Dr. Chang:

I am pleased to inform you that your manuscript has been deemed suitable for publication in PLOS ONE. Congratulations! Your manuscript is now with our production department. 

With kind regards,

on behalf of

Prof. Dong-Yan Jin 

Academic Editor

PLOS ONE